# Reduced Heating Wireless Energy Transmission System for Powering Implanted Circulatory Assist Devices: Benchtop and In-Vivo Studies

**DOI:** 10.3390/s25051311

**Published:** 2025-02-21

**Authors:** Mohammad L. Karim, Rachel Grimes, Harry Larkin, Antonio M. Bosnjak, James McLaughlin, Paul Crawford, David McEneaney, Omar J. Escalona

**Affiliations:** 1Nanotechnology & Integrated BioEngineering Centre (NIBEC), School of Engineering, Ulster University, Belfast BT15 1AP, UK; m.karim@ulster.ac.uk (M.L.K.);; 2Galvani TECH Ltd., School of Engineering, Ulster University, Belfast BT15 1AP, UK; 3Paul Crawford Veterinary Services, Larne BT40 3RW, UK; paul@paulmrcvs.com; 4Cardiovascular Research Unit, Craigavon Area Hospital, Portadown BT63 5QQ, UK

**Keywords:** left ventricular assist device (LVAD), rechargeable implanted medical battery, transcutaneous energy transmission system (TETS), wireless charging rate, tissue heating effects

## Abstract

This study aimed to develop a novel Transdermal Energy Transmission System (TETS) device that addresses the driveline complications faced by patients with advanced heart failure (HF). Our TETS device utilizes a two-channel configuration with a very-low duty cycle and a pulsed RF power transmission technique, along with elliptically shaped flexible coil inductive coupling elements. We integrated a battery charging controller module into the TETS, enabling it to recharge an implanted Lithium-Ion (Li-Ion) battery that powers low-power-rated Circulatory Assist Devices, or left ventricular assist devices (LVADs). Benchtop measurements demonstrated that the TETS delivered energy from the implanted coils to the battery charging module, at a charging rate of up to 2900 J/h, presented an average temperature increase (ΔT) of 3 °C. We conducted in vivo measurements using four porcine models followed by histopathological analysis of the skin tissue in the implanted coils areas. The thermal profile analysis from the in vivo measurements and the calculated charging rates from the current and voltage waveforms, in porcine models, indicated that the charging rate and temperature varied for each model. The maximum energy charging rate observed was 2200 J/h, with an average ΔT of 3 °C. The exposed skin tissue histopathological analysis results showed no evidence of tissue thermal damage in the in vivo measurements. These results demonstrate the feasibility of our developed TETS device for wireless driving implanted low-power-rated LVADs and Li-Ion charging.

## 1. Introduction

Cardiovascular diseases (CVDs) encompass various heart-related complications, one of which is advanced heart failure (HF). HF is a leading cause of death not only in Europe and the USA but also globally [1,2,3]. In 2022, nearly 702,880 people died from heart disease in the USA, which represents 1 in every 5 deaths [4,5]. Additionally, over 10 million people in Europe and the USA are living with advanced HF, with approximately 1.1 million new cases diagnosed each year. Furthermore, HF has a significant economic impact on healthcare systems [6]. Without intervention, patients with advanced heart failure (HF) face a grim prognosis, often worse than that of many cancers. For those with advanced HF—approximately 300,000 individuals in Europe and the US—cardiac transplantation might be a viable option. However, only about 9000 donor hearts are available worldwide each year. Patients awaiting a heart transplant can be supported by a circulatory assist device (CAD), or a left ventricular assist device (LVAD), which is a small electromechanical pump placed next to their own heart. While LVADs can be lifesaving, they do have a significant drawback [7,8,9,10]: their high-power (3–10 Watts) requirements necessitate a driveline that passes through the skin (percutaneously pierced) to connect to an external battery pack. This percutaneous driveline is prone to skin infections, which often lead to hospital admissions, antibiotic therapy, and sometimes surgical revisions. As a result, the quality of life for patients using LVADs can be greatly diminished [10].

To address driveline issues, wireless power-driven LVADs have been developed using inductively coupled Transdermal Energy Transmission System (TETS) [11]. Wireless power transmission (WPT) is the main feasible solution that can eliminate the use of a percutaneous driveline, which is the current clinical practice, by means of a wireless TETS device [12]. However, a significant challenge and technological solution paucity remains, that being the heating effect on the skin tissue surrounding the radiofrequency (RF) coupling elements of the TETS (external transmitter and implanted receiver coils), which can lead to localized skin damage, or patient discomfort at least. Furthermore, the tissue heating problem is the primary challenge to any solution development effort for a high-power-rated (>3 W) wireless TETS, to the point that, currently, no commercial TETS solution has sustainably taken off yet, in the nearly 70-years-long TETS development history [13].

The impact of heat on skin tissue is complex, and elevated temperatures during radio frequency (RF) power transmission can cause irreversible damage to the tissue [14,15]. TETS are designed based on the clinical necessity to continuously operate an LVAD for patients with advanced heart failure (HF). However, prolonged use of the TETS raises concerns about temperature management. There are specific temperature regulatory aspects concerning implanted devices. The FDA indicates that no special considerations are necessary for temperature increases of less than or equal 2 °C [16]. According to guidelines from the International Electrotechnical Commission (IEC), any implanted medical device or medical equipment that is in close contact with the skin should not exceed 43 °C [17]. Additionally, The International Organization for Standardization (ISO) also specifies the thermal regulation of an implantable medical device. The temperature increase (ΔT) in an implanted medical device should not exceed 2 °C, considering a 37 °C base body temperature [18]. These regulations must be adhered to by the TETS device to ensure safety and compliance.

In our previous study, we developed and reported a TETS device for Medical Implants (MI) as an innovative approach towards a wireless power solution for LVADs [11,19]. Our research, including the in vivo measurements results, has demonstrated the effectiveness of TETS in wirelessly transferring sufficiently high power for LVADs with reduced thermal effects associated with inductively coupled RF power energy transfer. Our novel TETS features thin, compact, flexible, and biocompatible flat elliptical spiral coil elements designed for high-power wireless applications. It operates through resonant inductive coupling, minimizing weak electromagnetic (EM) radiation and addressing tissue heating effects. Our TETS concept uses high-energy RF energy pulses transmitted over a relatively short time interval, followed by an idle period, to allow tissue temperatures to decrease through capillary actions, such as blood perfusion around the implanted coils. The system design has been developed specifically with features that enable cooling by utilizing tissue blood flow adjacent to the subcutaneous muscle. Additionally, our proprietary pulsatile charging protocol minimizes tissue heating and eliminates the need for a driveline [19,20,21,22]. Furthermore, in our approach for this study, considerations were given to recently reported evidence of an enhanced Li-Ion battery charging process when implemented in pulsed mode [23], rather than the conventional continuous charging mode. Our TETS employs an implantable rechargeable battery to power an LVAD which operates at a power rating of less than 5 Watts. During a certain period of the day, the LVAD would be powered by the implanted battery in standalone mode during a window time of about 4 h, granting patients the freedom to perform daily activities without restrictions. During the rest of the day and at night, patients will wear a charging vest equipped with an external battery and wireless transmission coils to charge the implanted battery.

This study includes a thermal analysis conducted under FDA-compliant conditions [14], along with benchtop, in vivo, and histopathological assessments of skin tissue. We aimed to charge a battery at a rate of 3000 J/h, for 20 h; for keeping reliable low tissue heating effects, and achieving an estimated total battery energy charge of 60,000 J. Clinical evidence shows that this would be sufficient to drive a HeartMate™ LVAD, Abbott Cardiovascular, Plymouth, MN, USA (4.6 Watts; based on a clinical evidence case) for about four hours, without any external energy supply transmission.

## 2. Materials and Methods

This section describes a two-channel inductive coupling Transcutaneous Energy Transfer System (TETS) device equipped with a battery charging module for charging a rechargeable implantable battery. The TETS device operates in a pulsatile transmission mode, utilizing varying voltage levels, pulse widths (PW), and idle times (IT) [11]—periods without transmission—to evaluate tissue heating effects and transmit power through the skin to charge the implantable battery. Figure 1 illustrates the two-channel inductive coupling TETS device used in benchtop and in vivo measurements.

### 2.1. TETS

It is widely understood, and intuitively accepted, that energy transmission efficiency of TETS is strongly dependent on the coupling coils plane (disc) separation gap, and also on their plane central axis alignment. The latter one is always ensured/set to be in perfect alignment by default. Hence, TETS energy transfer efficiency is usually studied for a range of coils separation distance, to characterize a TETS device. For the particular TETS device used here, the respective energy transfer efficiency was previously characterized and reported [11]. There, the adopted efficiency definition is the DC-to-DC energy transfer efficiency, which includes the energy loses in the adopted Class E, resonant inductive coupling RF power transmitter amplifier methods [24], and loses in the resistive component of the coupling coils at 200 kHz (measured to be about 2.5 Ω, at 200 kHz, for the coil elements used), indicating that for a 3 to 6 mm gap range, an average D-to-DC energy transfer efficiency of 90% was evaluated [11].

In this study, a two-channel inductive coupling TETS prototype, presenting an energy transfer efficiency as indicated above (90%), was developed to conduct preclinical studies using four porcine models for in vivo measurements. Figure 1 illustrates a schematic block diagram of the TETS device. The TETS employs various transmitter (Tx) supply voltage levels, pulse widths, and idle times to facilitate benchtop studies and develop preclinical trial protocols. Table 1 outlines the pulse widths, idle times, and voltage levels used in the benchtop model for designing the pulsed transmission protocol to conduct the preclinical study. The TETS architecture features two channels, each with a primary and secondary coil.

Figure 1 shows the inductive coupling elements (coils), with two transmitter (Tx) units and two receiver (Rx) units. The device operates at a resonant frequency of approximately 200 kHz. Our TEST system was developed using specific voltage, pulse width, and idle time parameters, as shown in Table 1. The proof of concept detailing these parameters has been reported in previous works [11,19]. This setup generates high-energy pulses over a short RF power transmission interval, followed by a longer idle cooling period for the tissue (no RF transmission), due to capillary action and blood perfusion. It operates with minimal duty cycles, ranging from 0.1% to 10%. Figure 2 presents illustrative photos of the actual prototype version of the TETS device in a bench setting, with Transmitter (Tx) module and its variable Voltage Supply unit, the Receiver module (Rx), which contains the rectifier, the supercapacitor bank and the Li-Ion battery charger controller, and the two channels coupling coils between the Tx and Rx modules.

### 2.2. Configurations of the Probes (Tx and Rx), Thermocouples and NTC Thermistors

Each probe (Tx and Rx) consists of four layers (each layer is 50 µm thick; altogether 200 µm thick) of flexi elliptically shaped coils. The coils’ detail can be found in our previous work [19]. Each probe is surrounded by a silicone layer (2–3 mm thick) to prevent direct coil contact, thus reducing the thermal effect while transmission of power occurs between the coils. However, the primary and secondary probe configuration are slightly different, the silicone layer is 3 mm thick, on the adjacent subcutaneous muscle side, in the secondary probe. Both the implanted and external probes are placed inside a transparent, thin (0.1 mm) polyethylene sheath, so the probes are not directly in contact with the body fluids. The probes’ configurations are illustrated in Figure 3, with six adhered negative temperature coefficient (NTC) thermistor sensors, and one centrally adhered thermocouple sensor on each probe coil side (primary and secondary). The thermocouples are used to monitor the real time temperature of the probes. The secondary probes (for channel 1 and channel 2; see Figure 1) are implanted approximately 3 mm underneath the skin tissue, in the 50 kg porcine model. The primary probes are fixed directly on the skin surface and aligned with the secondary implanted probes.

The NTC thermistors acquire voltage signals. The following equation converts the recorded thermistor signals into temperature (T) in Kelvin (K).(1)$$\frac{1}{T}=\frac{1}{T_{0}}+\frac{1}{\beta }\ln\frac{R_{m}}{R_{0}}$$
where$$R_{m}=\frac{R_{0}}{\left(\frac{v_{ref}}{v_{m}}-1\right)}$$$$R_{0}=10\;k\Omega$$$$T_{0}=25\;{}^{\circ}C$$$$v_{ref}=2.048\;V$$$$V_{m}=NTC\;signal\;reading\;as\;a\;voltage$$β=temperature coefficient of the thermistor

### 2.3. Benchtop Model

Benchtop in vitro models were developed with a water thermal perfusion microcirculation emulation subsystem to assess the functionality and effectiveness of TETS, transmitter (Tx) and receiver (Rx) coil probes and identify the best protocol for the in vivo preclinical trial. A series of benchtop trials were completed as part of the studies. The details of these benchtop tests are outlined below. The benchtop models were designed to simulate the environment presented in the porcine model trial, using a translucid polyethylene water container (see illustration in Figure 4), and water flow to emulate blood’s circulation effects on temperature regulation surrounding the same coil probe to be implanted in the in vivo tranche of study. The benchtop polyethylene rectangular cuboid shaped container was thermally insulated with polystyrene cladding; to help maintain a stable water temperature. A crucial and integral component of this model was the thermostatically controlled water heater. A 500 W LCD thermostat-controlled heater was used for regulating water temperature in the container. It automatically turned on when the temperature fell below 34 °C, bringing it back up to a level close to a porcine model normal body temperature, around 34–36 °C. The setup, including the water container, heater and water circulation pump, is illustrated in Figure 4.

The secondary coils of Channel 1 and Channel 2, with their polyethylene sheath, were submerged in the water tank. The externally located primary coils were then aligned with the secondary coils and adhered to the outer side wall of the translucid water container. The separation gap between the primary and secondary coil was set to 6 mm, including the water container wall thickness (1 mm).

### 2.4. Preclinical Study (In Vivo Measurements)

The in vivo measurements were conducted on four pigs under general anesthesia (average weight: 50 kg; average body temperature: 36–37 °C; X male and Y female) under consistent measurement conditions. These conditions included the same power levels, pulse width, and idle time (no transmission), following a protocol developed from the data obtained from the benchtop model assessments described above. This study received ethical approval from the Agri-Food and Bioscience Institute’s (AFBI) Animal Welfare & Ethical Review Board. A project license (PPL 2900), validity dated from Mar/2021 to Feb/2026, was obtained under the Animals (Scientific Procedures) Act from the Northern Ireland Department of Health.

Each pig was sedated using a combination of morphine, midazolam, medetomidine, and ketamine administered through intramuscular injection. They were then transferred to the surgical facility, and a cannula was placed in an auricular vein. General anesthesia was induced with propofol administered intravenously until the desired effect was achieved, followed by the placement of a cuffed endotracheal tube (8–9.5 mm). General anesthesia was maintained with isoflurane in oxygen and medical air (FiO2 0.5). The pigs were ventilated to ensure normocapnia throughout the procedure (typically 18 breaths per minute with a tidal volume of 475 mL). Arterial blood pressure was monitored via a cannula inserted into a branch of the medial saphenous artery. Isotonic fluids were administered intravenously at a 5 mL/kg/h rate.

After skin preparation, two subcutaneous pouches were surgically created on each pig’s left and right sides: caudal to the elbow over the dorso-lateral thoracic wall. The pig was placed dorsal recumbency (Figure 5). At the end of the anesthetic period, the pigs were euthanized without recovering from anesthesia, by intravenous administration of an overdose of a barbiturate.

### 2.5. Tissue Samples Histopathology Analysis

Tissue samples of skin from the areas between the implanted receiving coils of both channels and adjacent to both implanted probes were taken postmortem along with a control tissue sample from a remote area, unaffected by surgery or energy transmission, for assessing any tissue damage due to TETS heating effects. Histology slides were prepared from the tissue samples. Figure 6 illustrates the labeling of finalized slides for histopathology analysis of the tissue samples. The tissue samples are reference labeled as follows: Control slide, Ch1 (A): adjacent tissue of the Channel 1; Ch1 (T): tissue sample between the primary and the secondary coils; Ch2 (A): adjacent tissue of the Channel 2; and Ch2 (T): tissue sample between the primary and the secondary coils.

## 3. Results

This section presents the thermal profiles’ data, the TETS output voltage, and current waveforms in the Li-Ion charging controller, to estimate the implantable battery energy charging rate (J/h) measurements, obtained from both the in vitro benchtop trial assessments and the in vivo trial measurements.

### 3.1. Benchtop Measurements

A series of benchtop assessments were conducted to optimize the charging rate of the implantable Li-Ion battery while operating at a trade-off solution for minimizing the heating effects of the implanted probes (coils) of the TETS device under development. The parameters, including the transmitter DC voltage supply level, RF power transmission pulse width (PW), and idle time (IT; no RF power transmission), are detailed in Table 1.

An initial baseline temperature was recorded for each benchtop model, prior to RF power transmission. Figure 7 illustrates the average baseline temperature prior to the benchtop study measurements; before any transmitted power. The initial temperature of the external probes was approximately 24 °C, while the implantable probes had an initial temperature of around 34 °C. Additionally, the container water temperature was consistently maintained at 34 ± 0.5 °C. The baseline temperature was recorded over a period of 20 min before conducting the benchtop in vitro study measurements, which were carried out for a duration of 1.5 to 2 h.

The protocol examined in this bench test involved a Tx supply voltage of 70 V, a transmission pulse width (PW) of 320 ms, and an idle time (IT) of 5 s. This particular parameters setting was empirically determined in order to maximize the energy transfer (or battery charging rate, J/h), at reduced skin tissue heating effects, below the tolerable temperature rise (ΔT) margin (2 °C), yet, yielding sufficient energy transfer when targeting to drive a LVAD power rate of 3.5 W. Figure 8 illustrates the benchtop measurements, and the thermal assessments of the implanted probes submerged in a water bath. Note that the surface temperature of the implantable probes (secondary side) began to rise as soon as the transmission started.

Channels 1 and 2 of the implanted probes each have two thermocouple positions on the surface. In Channel 1, “implanted (1)” refers to the core temperature, while “implanted (2)” represents the temperature outside the coil. Similarly, for Channel 2, “implanted (1)” indicates the core temperature, and “implanted (2)” indicates the outside temperature of the coil. The configuration of the probes was as illustrated in Figure 3.

During this protocol, the core temperature of Channel 1 increased from approximately 34.5 °C to 37.5 °C (ΔT = 3 °C). Channel 2 also experienced a temperature rise of 3 °C (ΔT = 3 °C) from its baseline. However, the temperature recorded at position 2 for both Channel 1 and Channel 2 showed a slight increase in the temperature and then stabilized. The silicone buffer layers above and below the coils are crucial in reducing direct heat transfer generated from the coil surface. Moreover, the microcirculation of the water was found to play a vital role for dissipating heat generated from the secondary coil. Figure 8a indicates that the core temperature at position 1, for both secondary coils, stabilized after approximately 800 s and maintained that temperature, 37.5 °C until the transmission was turned off. Once the transmission ceased, the temperatures at positions 1 and 2 decreased immediately. In Figure 8a, we can see a decrease in temperature just before 6000 s. This is during the period of the test with no transmission. With the rapid decrease experienced by both implanted probes we can assume that the water bath temperature fell below 34 °C at this stage. This is due to the cooling profile of each probe being recorded after transmission. During this process, the water bath temperature was not maintained as rigorously as before, the submerged heater is turned off, and the system cools down to room temperature (the average room temperature was 22° C).

Figure 8b,c display the TETS output current and voltage waveforms into the battery charging controller, as recorded by a digital oscilloscope, for the protocol (70 V, 320 ms, and 5 s) used in this benchtop study measurements. The total energy transferred to recharge the implantable battery was 2900 J, over one hour of power transmission for both Channel 1 and Channel 2. Therefore, the battery charging rate in this protocol was approximately 2900 J/h. This rate was calculated from the oscilloscope’s recorded current and voltage simultaneous waveforms capture data, at the input of the battery charging module. The benchtop results demonstrated a temperature increase of 3 °C (ΔT = 3 °C), associated with a trade-off battery recharge rate of 2900 J/h, which is sufficient to stand alone power a 3.5 W rated LVADs for about 4 h without any external power supply, battery modules, or driveline, after a Li-Ion charging period of about 18 h.

### 3.2. In Vivo Temperature Measurements

The in vivo measurements were conducted using the protocol that had been tested in benchtop measurements. During the two-hour in vivo study, we recorded the temperature profile, current, and voltage signals from the battery charging module in both Channel 1 and Channel 2, including data from the external (primary) and implanted (secondary) probes. The in vivo measurements involved a model with four porcine subjects under consistent experimental conditions. Initially, we recorded a baseline temperature for all porcine models without power transmission. This allowed for a stabilization period of 20–25 min to achieve a steady body temperature in the porcine models. The baseline temperatures measured are presented in Figure 9.

Six NTC thermistors adhered to the surface of the implanted coil and recorded the blood temperature when no power was transmitted. The recorded signals were then converted into degrees Celsius (°C) temperature readings. It is important to note that the temperature varied depending on the position of the thermistors. Figure 9a,b illustrate the baseline temperatures of the implanted coils for Channel 1 and Channel 2, respectively. Meanwhile, Figure 10a,b show the averaged baseline temperatures obtained from the six NTC thermistors for both Channel 1 and Channel 2, including both external and implanted sides.

Channel 1 exhibited a slightly higher temperature (approximately 0.5 °C) than Channel 2 on the implanted sides; however, the external temperature of Channel 1 was 1.5 °C higher than that of Channel 2. Both channels (external and implanted) reached a stable plateau following the stabilization period. Establishing the baseline temperature is critical for assessing the actual temperature rise (ΔT) during power transmission.

At the beginning of the in vivo measurements, the applied protocol was the same as that used in the benchtop measurements. However, during the in vivo measurements, the real-time temperature monitoring via thermocouples indicated temperature rises (ΔT) of more than 5 °C. Consequently, the protocol was revised: the voltage level was reduced to 50 V, while the pulse width remained at 320 ms, and the idle time stayed at 5 s, consistent with the benchtop protocol. This revised protocol effectively reduced the temperature increase, limiting it to a maximum ΔT of 3 °C. Figure 11 shows an in vivo average heating and cooling profile of a porcine model (model 1). For Channel 1, the average temperature of six NTC thermistors rises (ΔT) up to 3.1 °C, and for Channel 2, the average temperature of six NTC thermistors rises (ΔT) up to 2.9 °C. As soon as transmission terminated, the tissue temperature started to drop. Note that the in vivo porcine model empirical ΔT average peak temperature resulting figure of 3 °C (between the two channels), should be interpreted as an indicator that, under the pulsed RF power operating transmission protocol, maximum possible battery charging rate (J/h) has been reached. Nevertheless, it is just 1 °C above the tolerable tissue temperature increase limit (2 °C), which could be easily manageable by fine-tuning reducing the transmitter (Tx) voltage supply level.

Notably, porcine models 1 and 2 exhibited higher temperatures in Channel 1 compared to Channel 2. Conversely, porcine models 3 and 4 showed higher temperatures in Channel 2 than in Channel 1. The average temperature rise was (ΔT) 3 °C in models 1 and 2, while models 3 and 4 experienced a slightly higher average temperature rise of (ΔT) 3.15 °C. Figure 12 illustrates the increased temperature (ΔT) for the four porcine models after subtracting the baseline temperature from the actual temperature of each model.

The charging rates for the four porcine models were estimated by analyzing the recorded current and voltage signals from the battery charging module. All signals were captured using an oscilloscope, and the data were post-processed to calculate the charging rates for each model. The results are presented in Figure 13. The porcine models 3 and 4 exhibited higher charging rates, exceeding 2000 J/h, while models 1 and 2 had charging rates of less than 2000 J/h. This measurement indicates that the temperature and charging rates vary between models.

All tissue samples’ histopathology analyses were negative; no tissue damage was detected as the result of heating effects, or exposure to any intensive RF electromagnetic fields within the inductive coupling area.

## 4. Discussion

The benchtop measurement results indicated that the selected protocol (70 V, 320 ms, and 5 s) achieved the maximum charging rate of 2900 J/h with the developed Transdermal Energy Transfer System (TETS), including the probes configuration, gap, and inductive coupling elements. This charging rate allows for recharging an implantable battery, enabling patients to disconnect from the externally wearable TETS transmitter subsystem vest for 4–5 h without needing an external power supply or battery modules, assuming a 3.5 W rated LVAD, and after an implanted Li-Ion battery charging time of about 18 h.

Additionally, it was observed that the overall temperature increase in the benchtop model was ΔT = 3 °C. This temperature profile demonstrates minimal heating effects from the selected protocol while achieving the maximum charging rate. The benchtop model also simulates the blood circulation thermal perfusion cooling factor, which is essential for dissipating the heat generated during power transmission, as discussed in our previous work [21]. In this setup, we utilized a high-energy pulsed transmission protocol; thus, when transmission is active, the temperature increases momentarily, inducing an increase in blood thermal diffusion process, which is kept for various seconds during the idle time phase, when there is no transmission, causing an enhanced cooling effect, in comparison with non-pulsed (conventional, continuous, low-energy) transmission. Thus, pulsed transmission offers potential advantages on effectively mitigating heating effects for wireless power-driven left ventricular assist devices (LVADs). At the same time, pulsed wireless charging protocols for Li-Ion batteries, as reported by Liu, et al. [23], offer a potential enhanced charging process (increased efficiency and faster charging); thus, supporting the importance of the study for future TETS developments in wireless charging of medically graded implantable Li-Ion batteries.

In the latter line of thought, the applications to new LVAD concepts of subcutaneous rotating magnet are of particular relevance for a possible wireless LVAD drive sustainable solutions [25,26], which require adopting a hybrid operation of main, long-term, direct blood pumping, which is supported by a separate channel for TETS implanted Li-Ion battery charging during some 12–20 h (a backup process), which would energize the main hybrid LVAD (rotating magnet type) via the internal battery, during a time that the external wireless energizing vest is taken off from the patient. Another application case is the new concept of miniature, intracardiac left atrial/ventricular pumps for treating heart failure patients with preserved ejection fraction [27], which are a low-power-rated (below 3 W) type of LVADs, and would require both an implanted Li-Ion battery and a TETS for wireless charging. These constitute state-of-the-art, current sustainable wireless powered LVAD solution applications.

The in vivo measurements in this study showed that the temperature and charging rate varied with each porcine model. Perhaps the maximum power transmission efficiency depends on the gap between the primary and secondary coils and the thickness of the skin tissue. However, healthy tissue is necessary to circulate heat via blood perfusion. We achieved a higher charging rate in the benchtop measurements, but when we applied the benchtop protocol, the tissue temperature increased dramatically. We reduced the voltage level to 50 V and conducted thermal and battery charging measurements. We were able to reduce the tissue temperature (ΔT = 3 °C); however, the maximum charging rate observed in porcine model no. 3 was 2200 J/h, which is 700 J lower than the benchtop measurements. The silicone buffer layers between, under and above of inductive coupling coils, played an important role in diffusing direct tissue heating effectively, thus avoiding any tissue damage. The histopathology analysis results confirmed that no thermal damage occurred as a result of transcutaneous RF power transmission between the coils. Further development work will be incorporated to increase the charging rate to drive higher-power-rated LVADs (above 3.5 W). The in vivo measurement data are required to effectively develop a complementary capacitive coupling in combination inductive coupling approach to wireless LVAD driving and battery charging system for medical implants (MI).

To further appreciate the value and unique contribution of this work, in contrast to other research teams, on addressing the current knowledge paucity for achieving an effective and robust solution to heating effect issues with TETS, we refer to the complementary investigation reported by Au et al. [14], on the thermal safety of a hermetically sealed TETS for energizing implanted mechanical circulatory support (MCS) devices, which are similar to LVADs, using ovine models. Their results indicated average implant surface temperatures rising to 38.31 °C, using a conventional continuous transmission mode (not pulsed transmission), and there is a knowledge paucity on the complementary and essential wireless Li-Ion battery charging performance associated to their TETS device; an implanted backup energy storage device is essential in any complete TETS solution [12]. In other various research teams and research approaches, Lucke et al. [28] implanted a silicone-encapsulated TETS in a porcine model, achieving power transfers of 6 to 12 watts, with a maximum implant temperature of 39.5 °C. In a novel approach, Horie et al. [25] present a new LVAD design that enhances patient safety and comfort by offering both an extracorporeal rotating magnetic system for wireless driving a subcutaneous heart pump, and is combined with a solely dedicated TETS channel for charging an implanted battery pack. This system allows for alternating active-state between two blood pumps tubed in series, ensuring continuous operation while keeping temperatures rise (ΔT) within the safe limit of 2 °C margin.

In contrast to conventional methods, our novel TETS device concept employs pulsed transmission protocol, with elliptical coils, achieving charging rates of 2000 J/h (in vivo) and 3000 J/h (bench-top) tests for 4.6 Watts-rated LVADs, with a maximum (peak) temperature just 1 °C above the safety margin at 3 °C. Furthermore, the tissue heating problem is the primary challenge to any solution development effort for a high-power-rated (>3 W) wireless TETS, to the point that, currently, no commercial TETS solution has sustainably taken off yet, in the nearly 70-years-long TETS development history [13].

## 5. Conclusions

We conducted investigations on Transdermal Energy Transfer System (TETS) through both benchtop and in vivo measurements to assess their heating effects while also recharging an implantable medical-grade battery for driving circulatory assist devices in general, such as left ventricular assist devices (LVADs), without using the percutaneous driveline in the current clinical practice. Other implantable medical devices of lower pawer rating expand the range of possible applications. Our TETS device employs a two-channel configuration and utilizes a pulsed radio frequency (RF) power transmission technique, along with an integrated battery charging controller module to recharge an implanted Li-Ion battery for standalone energizing a low-power LVADs.

Benchtop measurements, with optimized pulsed transmission protocol (70 V, 320 ms, and 5 s), demonstrated battery charging at a rate of 2900 J/h, with a maximum temperature increase (ΔT) of 3 °C above the baseline temperature. This charging capability enables the implantable battery to support a patient’s disconnection from the externally worn TETS transmitter vest for 4 to 5 h, according to the needs of the patient, without requiring an external power supply or battery module, assuming the LVAD has a power rating of 3.5 W, and after approximately 18 h of battery charging time, and considering that the amount of power required to maintain the function of the LVAD, while battery charging, may be provided by a separate dedicated TETS channel suitable to the LVAD electrical model, or using a different wireless energy supply approach, such as by rotating magnet, as discussed above. However, the in vivo measurements revealed that the temperature and charging rate varied with each porcine model. While we achieved a higher charging rate during benchtop testing, applying the same protocol in the in vivo resulted in significant tissue temperature increases. To mitigate this heating effect, a pulsed transmission protocol with a lower voltage level (50 V, 320 ms, and 5 s), helped to reduce the tissue temperature to an increase of 3 °C (ΔT = 3 °C), though it decreased the charging rate to 2200 J/h. Moving forward, further development will aim to enhance the charging rate, enabling wireless driving of higher-power-rated LVADs (above 3.5 W) while further reducing the tissue heating effects for our TETS system.

## 6. Patents

The Ulster University proprietary design spiral–elliptical coils and RF pulsed waveform Transcutaneous Energy Transfer Systems (US patent no. 11,191,973 B2, granted on07/Dec/2021; Canada patent no. 2993759; China patent no. 201680051039.3, granted on15/Apr/2022; European patent no. EP3328490, granted on 31 August 2022).

## Figures and Tables

**Figure 1 sensors-25-01311-f001:**
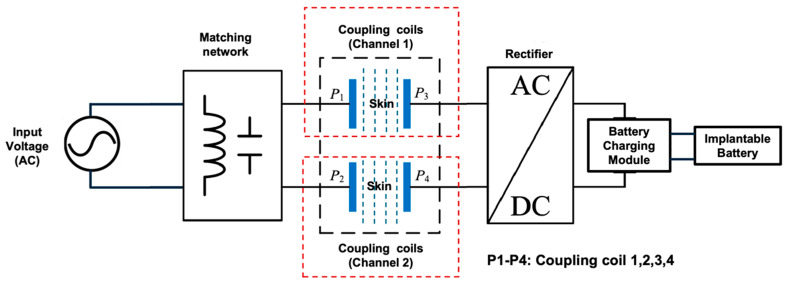
A schematic block diagram of a TETS device with inductive coupling elements and battery charging module.

**Figure 2 sensors-25-01311-f002:**
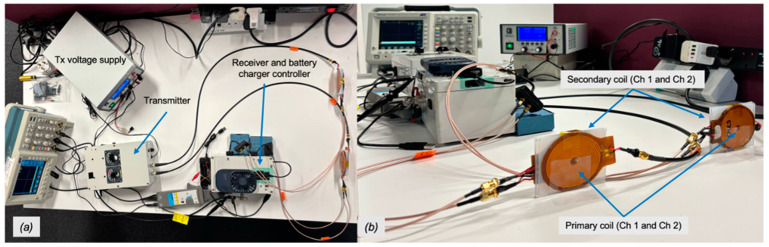
A prototype version of the 2-channel TETS with Tx, Rx modules and the inductive coupling coil elements in a bench test. (**a**) Top view photo of overall system. (**b**) Side view of coil elements.

**Figure 3 sensors-25-01311-f003:**
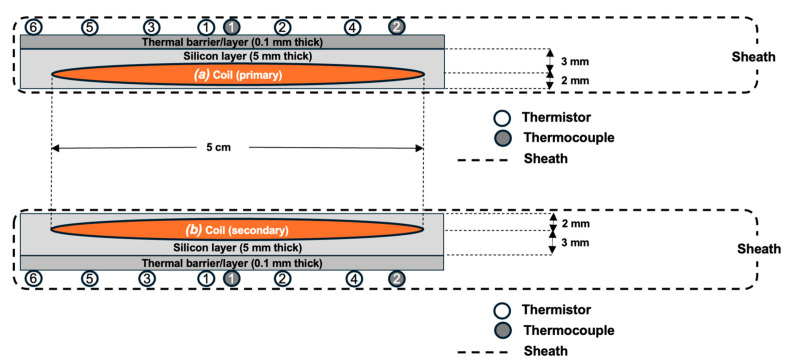
An illustration of the probes (Tx and Rx) configurations with NTC thermistors and thermocouples adhered to silicone thermal diffusion layer.

**Figure 4 sensors-25-01311-f004:**
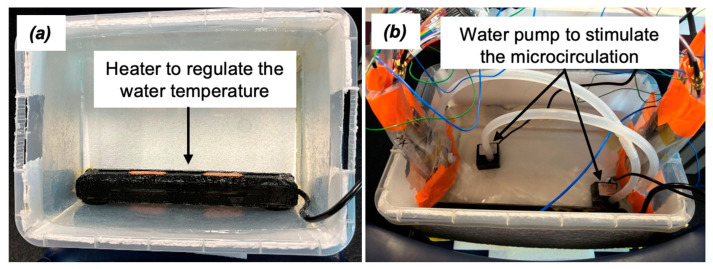
The benchtop setup, (**a**) a heater submerged under the water to regulate the water temperature, (**b**) water pumps to emulate the microcirculation of water through a 2nd sheath containing the secondary coil probe.

**Figure 5 sensors-25-01311-f005:**
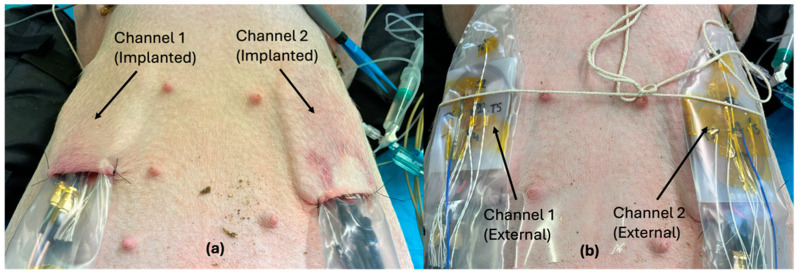
An in vivo measurement: (**a**) implanted coils placed underneath the skin pouches and (**b**) the external coils placed above the skin and aligned with implanted coils.

**Figure 6 sensors-25-01311-f006:**
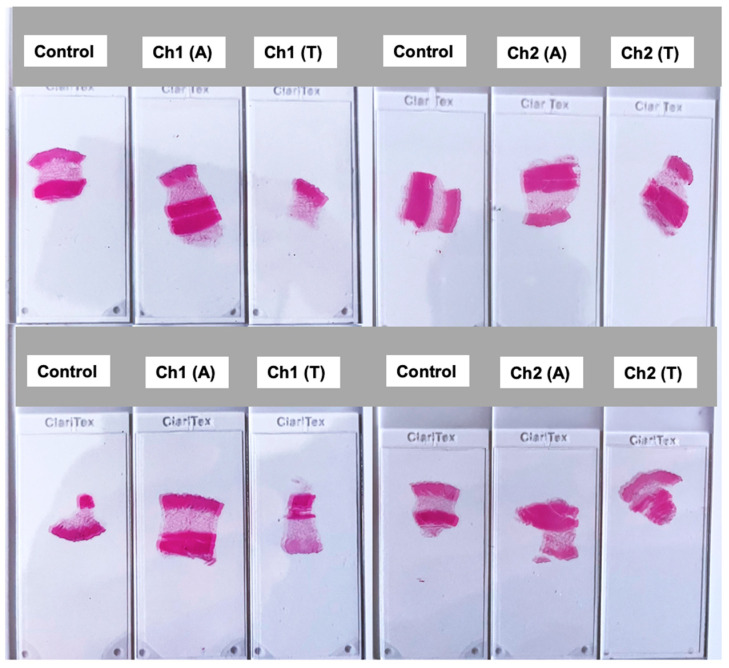
Histopathology slides labeling to investigate any tissue damage in the porcine model study.

**Figure 7 sensors-25-01311-f007:**
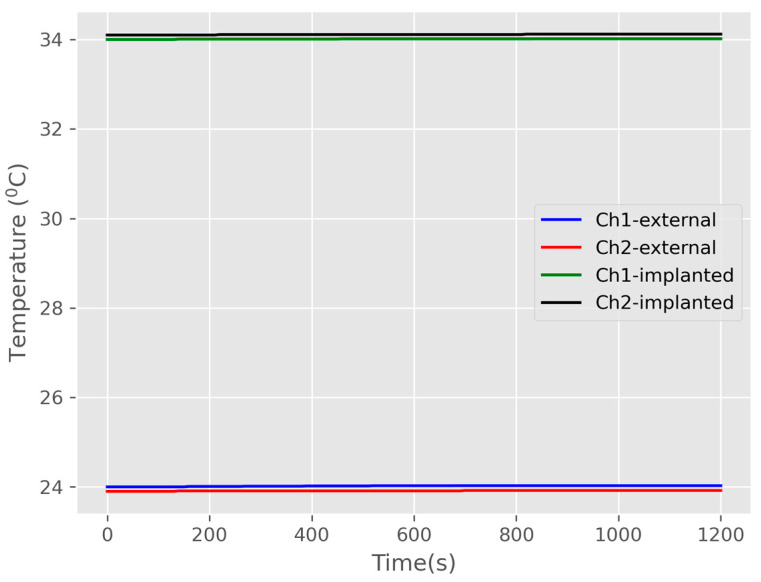
An initial 20 min baseline temperature of the external and the implanted probes (Channel 1 and Channel 2) without RF power transmission.

**Figure 8 sensors-25-01311-f008:**
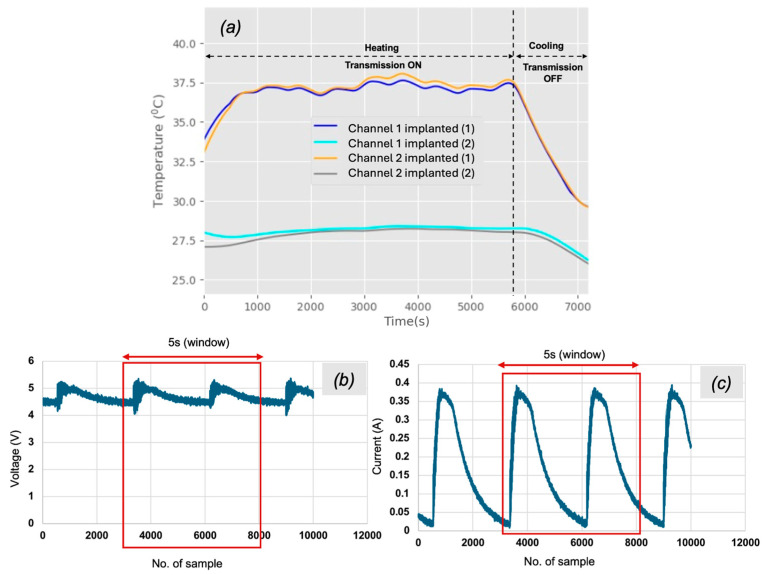
The benchtop measurements of (**a**) Channel 1 and Chennel 2 (implanted) heating and cooling profile, (**b**) the voltage waveform and (**c**) the current waveform of the selected protocol (70 V, 320 ms and 5 s).

**Figure 9 sensors-25-01311-f009:**
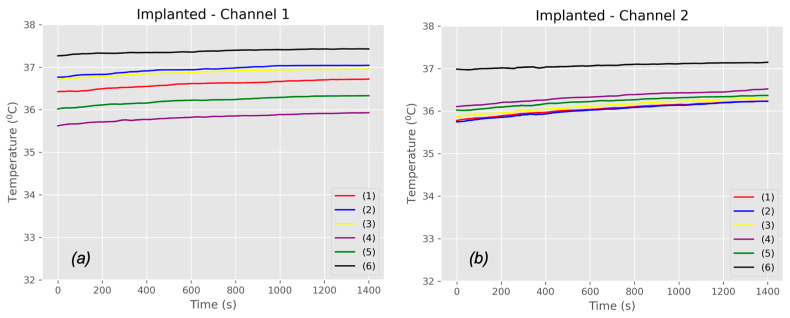
An in vivo baseline temperature measurement: (**a**) Channel 1—implanted, (**b**) Channel 2—implanted.

**Figure 10 sensors-25-01311-f010:**
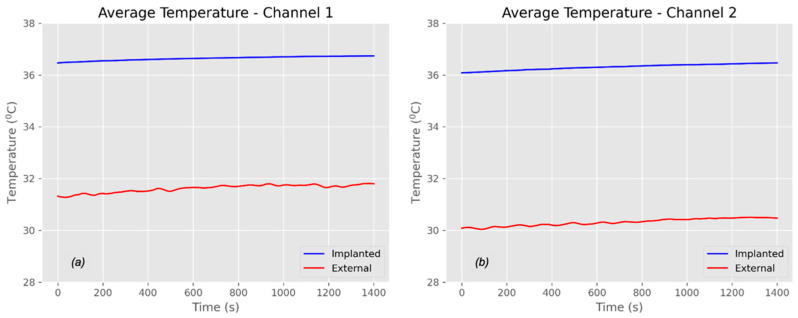
An in vivo average baseline temperature: (**a**) Channel 1—external and implanted and (**b**) Channel 2—external and implanted.

**Figure 11 sensors-25-01311-f011:**
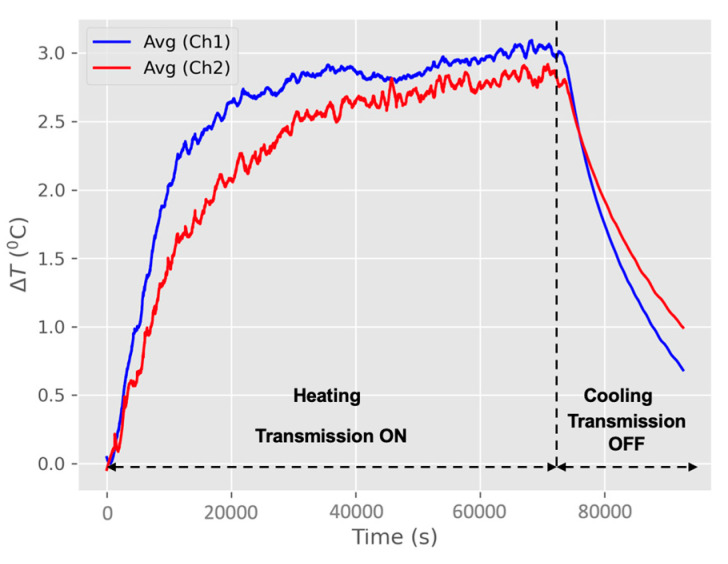
An in vivo average heating and cooling profile of model 1: (a) Channel 1—implanted and (b) Channel 2—implanted.

**Figure 12 sensors-25-01311-f012:**
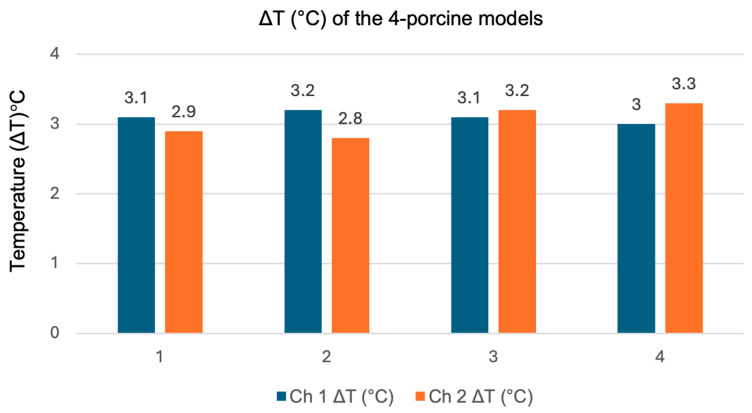
The temperature changes (ΔT) in the 4-porcine model in Channel 1 and Channel 2.

**Figure 13 sensors-25-01311-f013:**
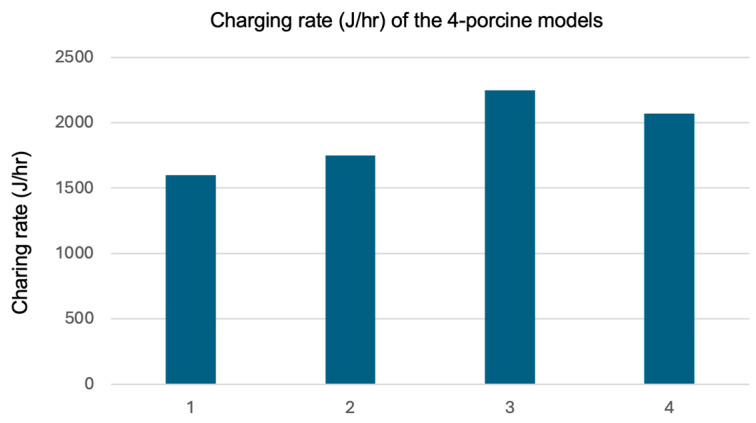
The calculated charging rate (J/h) of the 4 porcine models.

**Table 1 sensors-25-01311-t001:** Combination of pulses with idle time and various Tx voltage supply levels.

Pulse Width (ms) (ON)	Idle Time (s) (OFF)	Tx Voltage Supply (V)
30	5	50
60	10	60
90	20	70
160	40	80
320	80	90
480	120	100

## Data Availability

The data presented in this article are available on request from the corresponding author. However, the data are not publicly available due to confidentiality.

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
