# Peer review of "Reduced Heating Wireless Energy Transmission System for Powering Implanted Circulatory Assist Devices: Benchtop and In-Vivo Studies"

_sensors, 2025, doi:10.3390/s25051311_

Round 1
Reviewer 1 Report
Comments and Suggestions for Authors
- Hanging text at the start of page 4, underneath the table. Make sure to carefully proofread.
- In table 1, how are the three values related? Are these just arbitrary tested parameters or is there a meaningful relationship between them?
- Why did you use 70V / 320ms on / 5s idle for the benchtop test? Was there a downselection process to chose this set of parameters from the options in Table 1?
-Same question for the In Vivo study. Would like you to explain why this set of parameters was chosen, why multiple parameters weren’t tested. It seems more like you adjusted to make things work rather than systematically evaluated possible charging options
- In figure 7, it appears that after turning off power the baseline temperature dropped well below the 34C set point for your water bath. What is the explanation for this?
- Please include graphs similar to Figure 7 for the in vivo data showing heating over time. It would be good to show the time courses for the two protocols you tested (the “too hot” one and the one that gave acceptable heating limits) as a comparison
Author Response
Manuscript sensors-3447284 (Round-1) response to Reviewer 1 comments:
The authors are most grateful for kindly devoted time time on reviewing this manuscript.
Please find our itemised responses below and the corresponding revisions/corrections highlighted/in track changes in the re-submitted files.
Comments 1: Hanging text at the start of page 4, underneath the table. Make sure to carefully proofread.
Response 1: Thank you for detecting this detail.
The hanging text has been fixed (in page 4, line 162).
We have now carefully proofread the manuscript.
Comments 2: In table 1, how are the three values related? Are these just arbitrary tested parameters or is there a meaningful relationship between them?
Response 2: Thank you for your helpful observation for improving Table 1 interpretation to the readers. Our TEST device was developed using knob selected setting positions for discrete values of RF transmitter (Tx) voltage supply levels, transmission time pulse width (pulsed transmission TETS, and idle time parameters, as shown in Table 1. The proof of concept detailing these parameters has been published in our previous work [11, 19]. This TETS device approach generates high-energy pulses over a short interval, followed by a longer cooling period for the tissue due to capillary blood thermal perfusion. It operates with minimal duty cycles, ranging from 0.1% to 10% (this clarification text content has been included in the revised manuscript; before Table 1, in page 4, lines 165-173).
Comments 3: Why did you use 70V / 320ms on / 5s idle for the benchtop test? Was there a downselection process to chose this set of parameters from the options in Table 1?
Response 3: Thank you for this valuable questions.
We tested every possible combination of TX voltage supply level, pulse width, and idle time, as shown in Table 1, in order to maximise the energy transfer (or battery charging rate, J/hr) at reduced skin tissue heating effects, below the tolerable temperature rise (ΔT) margin of 2ËšC. Therefore, the 70V (Tx voltage), 320ms (p.w.) and 5s idle time transmission parameters set up combination presented sufficient energy transfer (charging rate) with reduced (within range) heating effect, in targeting to drive a 3.5 watts LAVD power rate. This clarification text was added to the revised manuscript in lines 334-338.
Comments 4: Same question for the In Vivo study. Would like you to explain why this set of parameters was chosen, why multiple parameters weren’t tested. It seems more like you adjusted to make things work rather than systematically evaluated possible charging options.
Response 4: Thank you for this helping comment to improve the manuscript.
As indicated above, in our Response 3, we tested all possible combinations in the bench-top measurements. Then, we chose the best combination to transfer sufficient energy and, in the meantime, minimise the tissue heating effects based on the bench results. The in-vivo measurements were conducted under time restrictions, due to the fixed anaesthesia period for a maximum of six hours. However, time includes surgery, implanting coils, and setting up all equipment. On average, we got a maximum four hours to measure the energy and temperature. For some cases (for the last 2 pigs), the measuring time was limited to only two hours (due to the lab’s ethical regulations). Also, we had to set the experimental protocol prior to the preclinical trail day.
Comments 5: In figure 7, it appears that after turning off power the baseline temperature dropped well below the 34Ëš C set point for your water bath. What is the explanation for this?
Response 5: Thank you for this request of clarification. In the original Figure 7, now is Figure 8 in the revised version, we turned off the transmission and the submerged water bath heater is also turned off, and the system cools down to room temperature (the average room temperature was 22°C); please see added lines in the revised version to clarify this point in page 9, lines 359-365).
Comments 6: Please include graphs similar to Figure 7 for the in vivo data showing heating over time. It would be good to show the time courses for the two protocols you tested (the “too hot” one and the one that gave acceptable heating limits) as a comparison.
Response 6: Thank you for your valuable suggestion. A new Figure (Fig.11) was generated similar to the former Figure 7 (now Fig.8). Figure 11 shows an in-vivo average heating and cooling profile of a porcine model (please see page 11).

Reviewer 2 Report
Comments and Suggestions for Authors
This paper presents a Transcutaneous Energy Transmission System (TETS) for wirelessly powering purposes. The manuscript must be reconsidered after major revision.
1) The current form of figures is more like a "patent" instead of a "paper". I strongly suggest the authors rebuild the figures (e.g., panel figure). For example, despite figure 1 & 2 shows the scheme of the device, a real-world photo is still necessary. The scheme of the device should focus on the underlying mechanism and composing components of the device.
2) To validate or emphasize the novelty of the work, comparison with existing work is necessary.
3) What is the energy transfer efficiency of the device?
4) A temperature increase of 3℃ is still too high for practical applications. What is the relation between power and temperature (increase), and value of a safe power?
Author Response
Manuscript sensors-3447284 (Round-1) response to Reviewer 2 comments:
The authors are most grateful for your devoted time on reviewing this manuscript.
Please find our itemised responses below and the corresponding revisions/corrections highlighted/in track changes in the re-submitted files.
Comments 1: The current form of figures is more like a "patent" instead of a "paper". I strongly suggest the authors rebuild the figures (e.g., panel figure). For example, despite figure 1 & 2 shows the scheme of the device, a real-world photo is still necessary. The scheme of the device should focus on the underlying mechanism and composing components of the device.
Response 1: The authors much appreciate the review concerns expressed by the reviewer here. The research team integrates academics and R&D staff from a TETS device development company (Galvani TECH Ltd, hence, the particular style reflected in some of the figures in this article. Under different circumstances, the team previously presented a TETS device architecture with an external transmitter (Tx), an implanted receiver (Rx), and sub-system components [see ref.19]. In response to the reviewer’s comments, we have inserted a new complementary image (Figure 2) of the actual prototype TETS device in a bench setting, with Transmitter (Tx) module and its variable Voltage Supply unit, the Receiver module (Rx), which contains the rectifier, the supercapacitor bank and the Li-Ion battery charger controller, and the two channels coupling coils between the Tx and Rx modules (see added text in lines 145-155 and Fig.2 at the end of Section 2.1).
Comments 2: To validate or emphasize the novelty of the work, comparison with existing work is necessary.
Response 2: Thank you for noting this concern and advice. In response to this particular concern we have extended the Discussion Section with the following content (in lines 561-586):
“To further appreciate the value and unique contribution of this work, in contrast to other research teams, on addressing the current knowledge paucity for achieving an effective and robust solution to heating effect issues with TETS, we refer to the complementary investigation reported by Au et al. [14], on the thermal safety of a hermetically sealed TETS for energising implanted mechanical circulatory support (MCS) devices, which are similar to LVADs, using ovine models. Their results indicated average implant surface temperatures rising to 38.31°C, using a conventional continuous transmission mode (not pulsed transmission), and there is a knowledge paucity on the complementary and essential wireless Li-Ion battery charging performance associated to their TETS device; an implanted backup energy storage device is essential in any complete TETS solution [12]. In other various research teams and research approaches, Lucke et al. [28] implanted a silicone-encapsulated TETS in a porcine model, achieving power transfers of 6 to 12 watts, with a maximum implant temperature of 39.5°C. In a novel approach, Horie et al. [25] present a new LVAD design that enhances patient safety and comfort by offering both an extracorporeal rotating magnetic system for wireless driving a subcutaneous heart-pump and is combined with a solely dedicated TETS channel for charging an implanted battery pack. This system allows for alternating active-state between two blood pumps tubed in series, ensuring continuous operation while keeping temperatures rise (ΔT) within the safe limit of 2°C margin.
In contrast to conventional methods, our novel TETS device concept employs pulsed transmission protocol, with elliptical coils, achieving charging rates of 2000 J/hr (in-vivo) and 3000 J/hr (bench-top) tests for 4.6 Watts rated LVADs, with a maximum (peak) temperature just 1°C above the safety margin at 3°C. Furthermore, the tissue heating problem is the primary challenge to any solution development effort for a high power-rated (> 3 W) wireless TETS, to the point that, currently, no commercial TETS solution has sustainably taken off yet, in the nearly 70-yearlong TETS development history [13]”.
Comments 3: What is the energy transfer efficiency of the device?
Response 3: Thank you for indicating this clarification point, in helping to improve the manuscript value to the broad readers. To address the potential question by readers, as flagged by this reviewer comment, we have added some clarifying text at the start of Section 2.1 (lines 121-131).
“It is widely understood, and intuitively accepted, that energy transmission efficiency of TETS is strongly dependent on the coupling coils plane (disc) separation gap, and also on their plane central axis alignment. The latter one is always ensured and set to be in perfect alignment by default. Hence, TETS energy transfer efficiency is usually studied for a range of coils separation distance, to characterise a TETS device. For the particular TETS device used here, the respective energy transfer efficiency was previously characterized and reported [11]. There, the adopted efficiency definition is the DC-to-DC efficiency, which includes the energy loses in the adopted Class E, resonant inductive coupling RF power transmitter amplifier methods [24], and loses in the resistive component of the coupling coils at 200 kHz (measured to be about 2.5 Ω, at 200 kHz), indicating that for a 3 to 6 mm gap range, an average D-to-DC energy transfer efficiency of 90% was evaluated [11].”
Furthermore, in this study, we focused on the charging rate for an implantable Li-Ion rechargeable battery. However, we must balance/limit the charging rate with the resulting skin tissue temperature increase. While we can enhance efficiency by implementing a high-voltage supply Tx setting, or with a wider transmission pulse width protocol, these would also raises the temperature. Our goal is to minimize tissue heating while maximizing the charging rate.
Comments 4: A temperature increase of 3℃ is still too high for practical applications. What is the relation between power and temperature (increase), and value of a safe power?
Response 4: Thank you for this valid observation, which many readers would similarly wonder about. Thus, we agree with this comment. As indicated above (in Response 3), a RF transmission power increase, by a fine increase of the TX voltage supply, will cause the tissue temperature to also rise. According to regulatory guidelines, the temperature of the implanted device should not exceed 2 °C above the body's normal temperature. This study was aimed to provide provides knowledge and understanding of the factors contributing to tissue heating effects, or tissue cooling by blood perfusion during the idle cycle of transmission, hence, achieving reduced heating effects. To address these reviewer’s comments, we clarify to the reader the correct interpretation of the 3 °C temperature result figure in the additional text in lines 464-469 (in Section 3.2), and in the extended last paragraph of the Discussion section (lines 607-613).

Round 2
Reviewer 2 Report
Comments and Suggestions for Authors
Thanks for revising the manuscript. Suggest accept.